# A Shared Cyber Threat Intelligence Solution for SMEs

Max van Haastrecht [1,*] , Guy Golpur [2], Gilad Tzismadia [2], Rolan Kab [2] , Cristian Priboi [3], Dumitru David [3], Adrian Răcătăian [3], Louis Baumgartner [4], Samuel Fricker [4,5], Jose Francisco Ruiz [6], Esteban Armas [6], Matthieu Brinkhuis [7] and Marco Spruit [1,7,8]

1 Leiden Institute of Advanced Computer Science (LIACS), Leiden University, Niels Bohrweg 1, 2333 CA Leiden, The Netherlands; m.r.spruit@lumc.nl
2 KPMG Somekh Chaikin, KPMG Millenium Tower 17 Ha'arba'a Street, Tel Aviv 6473921, Israel; ggolpur@kpmg.com (G.G.); gtzismadia@kpmg.com (G.T.); rkab@kpmg.com (R.K.)
3 CERT-RO, Italiană Street 22, 030167 Bucharest, Romania; cristian.priboi@cert.ro (C.P.); dumitru.david@cert.ro (D.D.); adrian.racataian@cert.ro (A.R.)
4 Institute for Interactive Technologies (IIT), University of Applied Sciences and Arts Northwestern Switzerland (FHNW), Bahnhofstrasse 6, 5210 Windisch, Switzerland; louis.baumgartner@fhnw.ch (L.B.); samuel.fricker@fhnw.ch (S.F.)
5 SERL-Sweden, Campus Gräsvik, Blekinge Institute of Technology, 371 79 Karlskrona, Sweden
6 Atos SA, Calle Albarracin 25, 28037 Madrid, Spain; josemistra@gmail.com (J.F.R.); esteban.armas.external@atos.net (E.A.)
7 Department of Information and Computing Sciences, Utrecht University, Princetonplein 5, 3584 CC Utrecht, The Netherlands; m.j.s.brinkhuis@uu.nl
8 Department of Public Health and Primary Care, Leiden University Medical Center (LUMC), Albinusdreef 2, 2333 ZA Leiden, The Netherlands
* Correspondence: m.a.n.van.haastrecht@liacs.leidenuniv.nl

**Abstract:** Small- and medium-sized enterprises (SMEs) frequently experience cyberattacks, but often do not have the means to counter these attacks. Therefore, cybersecurity researchers and practitioners need to aid SMEs in their defence against cyber threats. Research has shown that SMEs require solutions that are automated and adapted to their context. In recent years, we have seen a surge in initiatives to share cyber threat intelligence (CTI) to improve collective cybersecurity resilience. Shared CTI has the potential to answer the SME call for automated and adaptable solutions. Sadly, as we demonstrate in this paper, current shared intelligence approaches scarcely address SME needs. We must investigate how shared CTI can be used to improve SME cybersecurity resilience. In this paper, we tackle this challenge using a systematic review to discover current state-of-the-art approaches to using shared CTI. We find that threat intelligence sharing platforms such as MISP have the potential to address SME needs, provided that the shared intelligence is turned into actionable insights. Based on this observation, we developed a prototype application that processes MISP data automatically, prioritises cybersecurity threats for SMEs, and provides SMEs with actionable recommendations tailored to their context. Subsequent evaluations in operational environments will help to improve our application, such that SMEs are enabled to thwart cyberattacks in future.

**Keywords:** cybersecurity; cyber threat intelligence; information sharing; SME; MISP

## 1. Introduction

The cybersecurity threat landscape is diverse and dynamic, as witnessed by several recent supply chain attacks with worldwide impact [1,2]. Attack sophistication is increasing [3] and it is now widely accepted that even nation-states are actively involved in the most advanced and persistent threats [4]. Unsurprisingly, the trend of increased complexity in attacks is expected to continue in the future [5].

These observations stand in stark contrast to the situation of small- and medium-sized enterprises (SMEs), who lack the knowledge and resources to appropriately address any cybersecurity threats [6]; never mind advanced threats. SMEs require the help of their



external environment to deal with cybersecurity attacks since they do not have internally available expertise. This lack of internal expertise is a key reason SMEs require different cybersecurity solutions than larger enterprises [7].

In this sense, the maxim "a problem shared is a problem halved" is fitting in the SME context. It is this maxim that is the driving force behind information sharing in the cybersecurity community [3]. Sharing cybersecurity intelligence has long been recognised as a key ingredient in raising our collective cybersecurity resilience. Yet, until recently, efforts in this area were fragmented and unsuccessful [8], with many feeling the advantages to sharing data were outweighed by the disadvantages [9,10].

This changed with the introduction of standardised cybersecurity intelligence taxonomies [11–13] and intelligence sharing platforms [14,15]. Especially the sharing of threat [16–18] and incident [19] information gained acceptance and popularity. Privacy concerns still remain regarding the sharing of cybersecurity intelligence [20,21]. However, the focus has now shifted to finding solutions rather than simply detailing problems [22–24]. Exploiting the properties of blockchain for privacy preservation is an example of a novel and promising approach [25,26].

Recently, the use of advanced data analytics [27,28] and machine learning [29,30] techniques to extract further insights from shared intelligence has spurred on optimism regarding the future of cybersecurity information sharing. Nevertheless, the literature remains eerily silent regarding the use of shared incident data to support SMEs; a group in dire need of help from their external environment.

SMEs have their own concerns regarding information sharing [21], and certainly require different treatments and solutions than other enterprise types [31]. This is perhaps most true for the least digitally mature SME categories: start-ups and digitally dependent SMEs. Along with the more mature digitally-based SMEs and digital enablers, the European DIGITAL SME Alliance [32] distinguishes these SME categories to emphasise that SMEs are not one homogeneous group, but rather a diverse set of businesses, with diverse needs.

SMEs require distinctly different solutions than other enterprises due to their lack of internally available cybersecurity knowledge and resources. Additionally, any solution looking to aid SMEs should recognise the heterogeneity within this group of enterprises. Based on the current trends in cybersecurity intelligence sharing literature, it is therefore unlikely that any of the prevailing approaches to using shared incident data are suitable for SMEs. Nevertheless, it can be expected that current approaches contain building blocks for useful SME approaches, especially due to the automatic nature of today's machine learning techniques.

Finding out how we can use shared cybersecurity information to aid SMEs is our main focus in this paper. Hence, we ask:

**RQ**: How can shared incident information be used to help improve SME cybersecurity?

We will answer our research question by first systematically reviewing current approaches to using shared incident data in Section 2. Here we will also provide a detailed analysis of the difficulties of using the VERIS Community Database (VCDB) [33] in the SME context. These efforts provide insight into what adaptations to current approaches are necessary to yield a useful solution for SMEs. Our review contributes to the literature by structuring insight into which types of shared cyber threat intelligence (CTI) approaches are suited to SMEs.

We then describe our proposed solution, which uses the Malware Information Sharing Platform (MISP) [14], in Section 3. Our proposed solution is the second main contribution of this paper. We cover three phases of our solution: the input (Section 3.1), the process (Section 3.2), and the output (Section 3.3). In Section 3.4, we provide a practical example of how our application helps SMEs, demonstrating its potential impact. Finally, we discuss our findings in Section 4 and conclude in Section 5.

## 2. Literature Review

Before proposing our methodology, we should investigate current approaches to using shared CTI. We conducted this investigation via a systematic literature review using the SYMBALS [34] methodology. SYMBALS combines the use of active learning techniques to speed up the title and abstract screening phase of a review, with backward snowballing to ensure adequate coverage of the literature.

Active learning is a machine learning technique where the algorithm discovers how to select the most relevant information to learn from. It has been successfully applied in systematic reviews, where it was shown to significantly reduce the number of screened papers while finding around 95% of relevant papers [35]. In this paper, we use the ASReview tool of van de Schoot et al. [36] in the active learning phase of SYMBALS.

Since pure active learning approaches tend to miss out on certain relevant papers, van Haastrecht et al. [34] proposed to complement the active learning phase with a backward snowballing phase. Backward snowballing refers to the practice of finding new relevant papers from the references of inclusions [37]. Hybrid methods, such as SYMBALS, have recently been shown to offer an ideal combination of speed and completeness in reviews [38]. This motivated our choice for the SYMBALS method.

We searched the Scopus database for the keywords presented in Table 1, where we restricted our search to conference and journal articles and English-language documents. Additionally, we focused on research published since 2016. In 2016, the Malware Information Sharing Platform (MISP) was introduced [14]. MISP is one of the most widely used threat sharing platforms, along with the Trusted Automated eXchange of Indicator Information (TAXII) [12]. Both MISP and TAXII facilitate information exchange using the Structured Threat Information eXpression (STIX) language [11], the de facto standard format for exchanging threat intelligence.

**Table 1.** Keywords and accompanying synonyms used in our search of the Scopus database.

| Keyword | Synonyms |
|---|---|
| cybersecurity | cyber security, information security |
| threat | event, attack, incident |
| sharing | share |

The choice to focus our review on the period since 2016 is no coincidence. Since the introduction of MISP, the subject matter of shared threat intelligence research has shifted. Whereas earlier research explored information sharing options [8,39] and outlined the barriers to sharing [9], research since 2016 has largely centred around how we can use shared intelligence.

Our database search yielded 546 results, of which 47 inclusions remained after applying the filtering steps of SYMBALS. The most common reason for exclusion was that a paper did not cover our topic of interest: the use of shared threat intelligence. This is not surprising, as the keywords we employed do not provide a guarantee of papers in our focus area.

We then proceeded to extract relevant data from our inclusions. One dimension we considered was the suitable organisation type for an approach. The European DIGITAL SME Alliance outlines four SME categories: start-ups, digitally dependent SMEs, digitally-based SMEs, and digital enablers [32]. The cybersecurity maturity of these SME categories progresses from the least mature start-ups to the most mature digital enablers [7].

Where start-ups are only beginning to realise the importance of cybersecurity, we can expect digital enablers to have embedded, automated cybersecurity processes [7]. Nevertheless, even digital enablers are unlikely to have the capacity to run a Security Operations Centre (SOC) which can monitor and analyse continuously gathered internal security intelligence. This is why we included a 'large enterprises' category to collect any

methods unsuited to any SME category. The first column of Table 2 depicts our considered enterprise categories.

Ramsdale et al. [40] offer a concise classification of CTI sources. They divide sources into internally sourced intelligence, externally sourced intelligence, and open-source intelligence. Internally sourced intelligence relates to data on events occurring within an organisation's IT infrastructure. External intelligence comes from structured threat intelligence feeds, such as those sourced from the TAXII and MISP platforms. Finally, open-source intelligence is defined as intelligence from publicly available sources such as news feeds and social media. We choose to not employ the commonly used abbreviation of open-source intelligence OSINT, as OSINT is more broadly associated with the methodology of collecting threat intelligence from publicly available sources.

Table 2 categorises our inclusions based on the suitability of their approach to different enterprise types and the type of intelligence source they build on. We should note that the enterprise categories of Table 2 are ordered by cybersecurity maturity. This means that if start-ups can use a particular approach, digitally dependent SMEs will automatically also be able to use that approach. Similarly, if an approach is classed as being suitable for digitally-based SMEs, it is not suitable for the less digitally mature start-ups and digitally dependent SMEs.

**Table 2.** The type of cyber threat intelligence used in each of our 47 inclusions, along with the minimum SME category maturity required to implement the proposed methodology.

| Category | External Intelligence | Open-Source Intelligence | Internal Intelligence |
|---|---|---|---|
| Start-ups | [41] | [42] | |
| Digitally dependent | | [43] | |
| Digitally based | [44,45] | [18,46–48] | [25,49] |
| Digital enablers | [50–52] | [53–57] | [26,58–67] |
| Large enterprises | [68–73] | [74,75] | [76–84] |

The first thing to notice about Table 2 is that very few of our inclusions specify shared CTI solutions suitable for start-ups and digitally dependent SMEs. We cannot expect these SMEs to collect and analyse internal intelligence, which explains why none of the internal intelligence approaches is suited to start-ups and digitally dependent SMEs. Internal intelligence approaches often require an internal security expert or even a SOC, which make them difficult to implement even for digitally-based SMEs and digital enablers.

Open-source intelligence methodologies often suffer from their open-ended nature, making them less actionable for SMEs. The collected data are often unstructured text and will generally only serve to inform the user, rather than assist them in concrete tasks. The two open-source approaches that are suited to less digitally mature SMEs have a very specific goal. In the first, the authors create a spam filter based on open-source spam data, which can then be used by organisations to prevent spam from reaching employee inboxes [42]. The second approach also uses publicly available spam data, but this time it is connected to organisation IPs and used as a tool to confront companies with their security level [43].

Although the mentioned open-source intelligence sharing methods have their merits for start-ups and digitally dependent SMEs, they only scratch the surface of what can be done to help SMEs. Structured external intelligence could be an outcome here, but, as Table 2 shows, most research is geared towards large enterprises. All of the external intelligence approaches for large enterprises use STIX as their data sharing format, and most use TAXII as the sharing platform. The benefit of STIX is that it is flexible and therefore facilitates many different indicators of compromise (IoCs). However, most research proposes methodologies whereby the STIX data are shared without much processing. This means the shared data retains much of STIX's complexity, and it is left to analysts at an organisation to interpret this data. SMEs simply do not have the resources for such activities.

The external intelligence approaches suited to SMEs still regularly employ STIX. However, they no longer use TAXII as a sharing platform, preferring less common platforms or a custom sharing platform. Approaches that apply a more extensive filtering process to provide organisations with concise insights are most suited to the least digitally mature SMEs. By comparing shared data to blacklists [44] or using the shared intelligence to advise on suitable production rules [45], digitally-based SMEs are aided in their detection process. However, detection is still a step too far for start-ups and digitally dependent SMEs, who are often still in the process of understanding their assets and attack surface [7].

The external intelligence approach suited to start-ups uses a feed of passwords identified in breaches to inform users of susceptible passwords [41]. As with the open-source intelligence approaches, it is the focused nature and clear aim of this approach that makes it accessible to all types of SMEs. The question remains whether we can go beyond these specific implementations while maintaining usability for the least digitally mature SMEs. Such solutions currently do not exist and would be immensely beneficial to SMEs.

We certainly believe it is possible to create such solutions. It is clear from our systematic review results that the solution lies in the use of structured external threat intelligence, preferably conforming to the STIX standard, which is sufficiently processed and filtered to yield actionable insights for SMEs. Section 3 explains our solution.

Before diving into our solution, it is worth investigating whether a similar approach using open-source intelligence would also be feasible. We noted earlier that one of the main issues with open-source intelligence for SMEs is its unstructured nature. However, structured open-source intelligence sources do exist. The VERIS Community Database (VCDB) [33] is commonly used in cybersecurity research [19,85] and also serves as the basis for Verizon's yearly Data Breach Investigations Report (DBIR) [86]. Altogether, VCDB seems to be the ideal CTI source.

As we look closer, however, problems start to emerge. VCDB is largely composed of data breach incidents collected by analysts from news reports. Although a data breach can be considered an outcome of a cybersecurity threat, it is more commonly classified as a type of threat. The European Union Agency for Cybersecurity (ENISA) is a prominent example of an institution classifying data breaches as a threat type.

ENISA publishes a yearly list of top threats and 'data breach' appears every year. Their ranking is constructed "through continuous analysis" of numerous "publicly available sources" [87]. Figure 1 shows a comparison of VCDB and ENISA threat rankings from 2012 to 2017. Of the 12 threats depicted, 11 appear in the ENISA top threats each year. The exception is the 'external environment threat' which was introduced by van Haastrecht et al. [88]. External environment threats comprise the threats resulting from third parties and suppliers interacting with an organisation. This threat category is especially relevant for SMEs, as we have seen in the proliferation of recent supply chain attacks [1,5]. Although ENISA has not included it in their top threats, the threats making up the external environment threats do appear in their overall threat taxonomy. For an overview of all threat definitions used in this paper, please consult Table A1 in Appendix A.

To produce Figure 1, we analysed confirmed SME incidents included in VCDB from 2012 to 2017, with 2017 being the most recent year for which confirmed incidents were available. VCDB can be seen as structured open-source intelligence, but it is based on unstructured open-source intelligence. The intermediate step of structuring the original data is a time-consuming task. Thus, a common drawback of structured open-source intelligence is that it is outdated by the time it becomes available. This is problematic when the cyber threat landscape is constantly changing.

VCDB defines small businesses as having fewer than 1000 employees, which is an exceedingly broad definition, given that it is more common to use 250 employees as the cut-off point for SMEs [89]. This curious SME definition is one of the reasons why using VCDB can be problematic in the SME context. Nevertheless, we persisted in our analysis and chose to use those incidents classified as involving companies with 100 or fewer employees. Yet, as can be observed from Figure 1, the rankings resulting from our VCDB analysis differ

from the ENISA rankings. Unsurprisingly, VCDB's focus on data breach incidents leads to a much higher ranking for the data breach threat. However, many of the other threats also have ranking progressions dissimilar to ENISA's rankings.

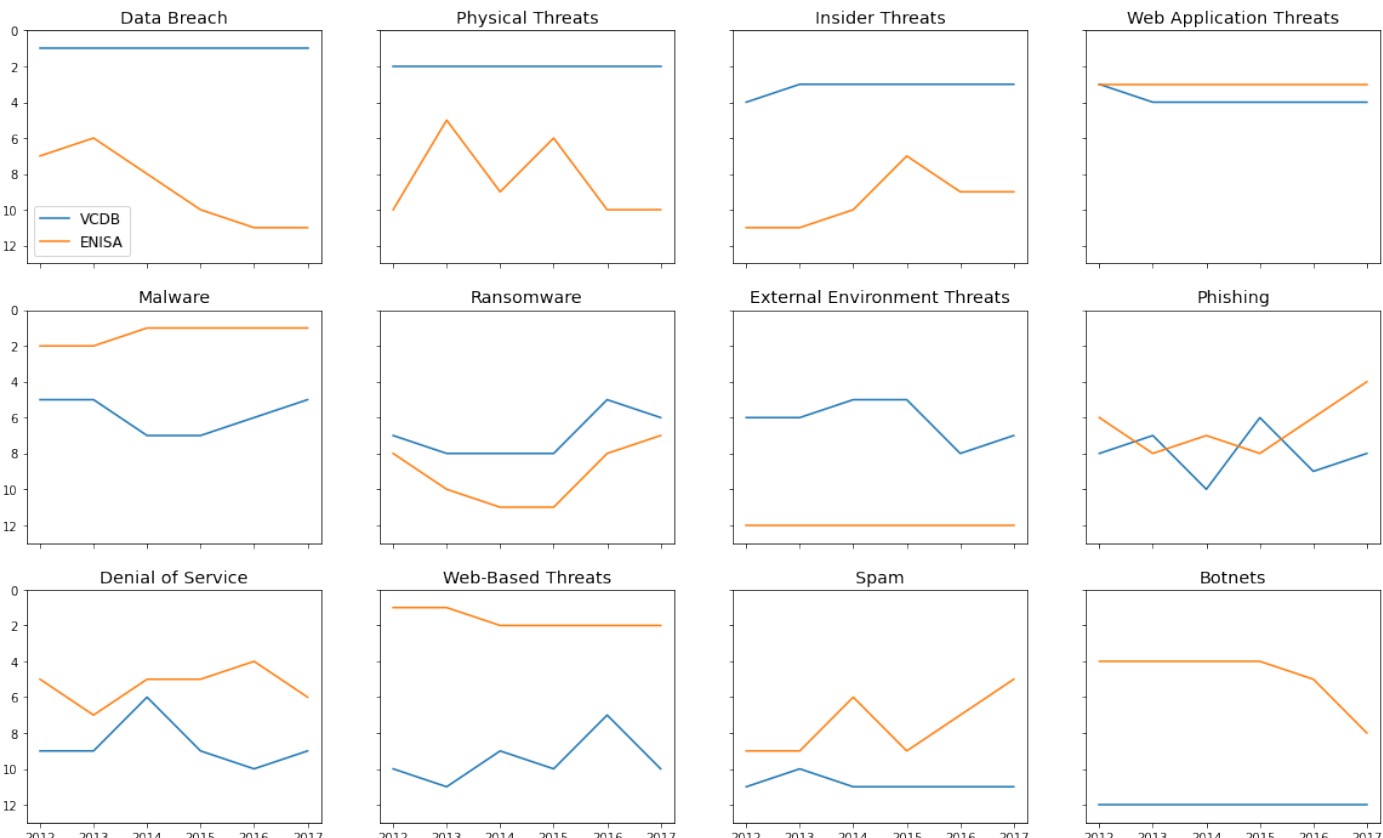

**Figure 1.** VCDB and ENISA threat rankings compared over time. For several threats we observe large ranking differences.

This points to two issues with using VCDB data. First, given the focus on data breach incidents, the data collected for VCDB is skewed heavily towards this threat type. This influences not only the data breach category but also all other categories, as threats that are highly correlated with data breaches will receive a higher ranking. Second, since the main collection method for VCDB incidents is the scanning of news reports, the threat ranking is biased towards newsworthy threat types. Data breach incidents often appear in the news, since in many countries there is an obligation to openly report such incidents. Phishing incidents, for example, are much less likely to be reported in news articles, as companies have no incentive to communicate their occurrence.

Further issues with VCDB relate to the fact that around 82% of the SME incidents originate from the US, that the English-speaking analysts collect almost exclusively English news articles, and that the manual process of its construction results in erroneously included incidents and duplicates. Altogether, this yields a VCDB threat ranking that is unlikely to reflect the ranking obtained when having perfect knowledge of incident frequencies.

Does that mean that the VCDB is useless to SMEs? No, certainly not. By being aware of the selection bias involved in constructing the VCDB, we can still use this data as input for the prioritisation of SME cybersecurity threats. We must take care to always complement VCDB information with other data sources, such as the ENISA rankings and expert assessments. With our approach, we hope to harness the beneficial aspects of VCDB, while taking care to avoid some of the traps associated with using its biased and outdated data.

## 3. Shared CTI Solution for SMEs

The European Horizon 2020 project GEIGER [90] aims to develop an adaptable, dynamic, and usable application to assess and improve the cybersecurity risk level of SMEs. The GEIGER application was designed with the existing body of cybersecurity knowledge in mind. However, the cyber landscape is not constant, meaning the GEIGER solution requires an updating mechanism. In this section, we will argue how shared CTI can be the key for continuously updating GEIGER.

Before turning to the solution we developed to improve the GEIGER application, let us recap what we have learned in the past two sections, to inform our solution design. We learned that SMEs generally lack the cybersecurity knowledge and resources to perform complex tasks [6]. Hence, they require understandable and actionable recommendations on how to improve their cybersecurity posture [88]. We learned that SMEs should not be seen as one homogeneous group, but rather as a heterogeneous set of enterprises with different characteristics and needs [7,32]. Any cybersecurity solution for SMEs should therefore be able to adapt based on SME characteristics, to provide tailored advice [31].

Lastly, any cybersecurity solution needs to be updated based on changes in the cyber threat landscape, which is where shared CTI enters the picture. Threat sharing platforms such as MISP ensure that "new threats can be identified more quickly" such that "response can be adequately coordinated" [91]. For larger enterprises, we may expect a security expert or SOC to be involved in this response process. However, such resources are rarely available at SMEs [7]. Therefore, our solution should incorporate an automated updating procedure facilitating adaptation to a changing threat landscape. We summarise our three requirements for an SME cybersecurity solution below:

1. The solution must provide understandable and actionable recommendations.
2. The solution should be able to adapt to different SME characteristics.
3. The solution should update automatically in response to a changing cyber threat landscape.

In the next sections, we describe how shared CTI could be the ideal prescription to meet the above requirements. The use of shared CTI involves an input, a process, and an output. We cover each of these elements in the context of the GEIGER solution, starting with the input: MISP data.

### 3.1. Input: Explaining MISP

The Malware Incident Sharing Platform (MISP) was introduced in 2016 [14] and has risen in popularity ever since. MISP is a flexible incident sharing platform that is compatible with STIX. The platform is supported by the Computer Incident Response Center Luxembourg (CIRCL), which explains why it is popular among many colleague Computer Emergency Response Teams (CERTs) across Europe.

MISP is a free and open-source platform for threat information sharing. MISP provides software for the sharing, storage, and correlation of IoCs related to cybersecurity incidents. The MISP data model is composed of events, which usually represent threats or incidents. Events, in turn, are composed of a list of attributes. Examples of attributes are IP addresses and domain names. Other data types exist in MISP, such as objects, which allow advanced combinations of attributes, and galaxies, which enable deeper analysis and categorisation of events.

MISP's data model is based on a JSON schema for event exchange, allowing for the classification of objects using different taxonomies. MISP comes with predefined taxonomies and users can define taxonomies according to their needs. Examples of included taxonomies are the ENISA and VERIS taxonomies. This allows CERTs to classify events according to their requirements, while still following accepted standards in the cybersecurity field. In Figure 2, we can see some examples of available taxonomies being used to classify incidents.

CERT-RO, the Romanian CERT that is a partner in the GEIGER project, uses MISP for the collection of cybersecurity alerts from different stakeholders. To comply with its legal

obligations, CERT-RO has developed a taxonomy for reporting specific events to Romanian cyberspace. All events from their sources and sensors use the CERT-RO taxonomy. The CERT-RO MISP implementation is based on the MISP module implemented in the National Cyber Security Platform (NCSP). This platform was developed to increase CERT-RO's technical capabilities related to cybersecurity incident management and information sharing. The platform is used for the collection, processing, and dissemination of data related to cybersecurity incidents, vulnerabilities, threats, events, and artefacts, including incident notifications received by CERT-RO. Information such as malicious URLs, IPs, and file signatures are usually distributed through this module.

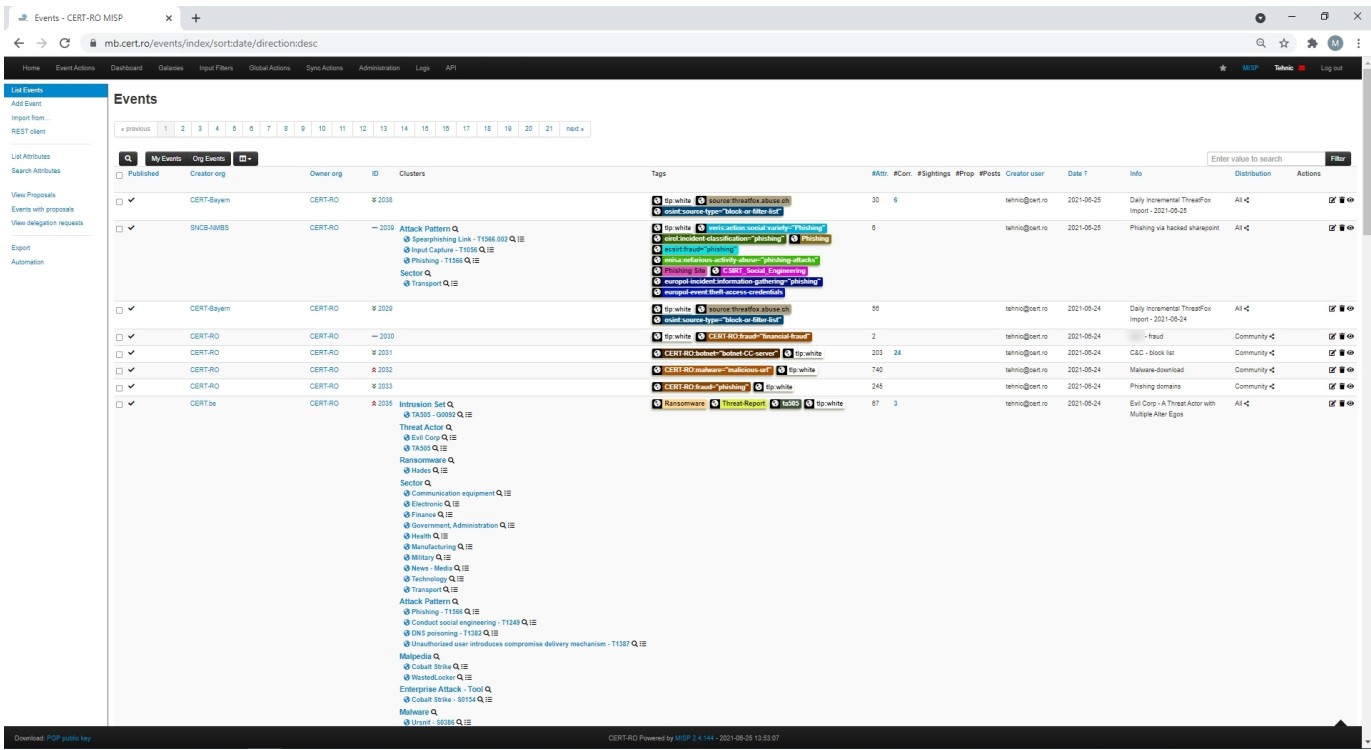

**Figure 2.** Examples of TLP:WHITE events that can be shared from CERT-RO's MISP instance to the GEIGER cloud.

CERT-RO's MISP data tagged with 'TLP:WHITE' is made available to GEIGER in a feed that can be imported in the GEIGER backend component in the cloud. TLP stands for Traffic Light Protocol; a protocol created to promote the sharing of information. TLP is a set of designations used to ensure that sensitive information is shared with the appropriate audience. It employs four colours to indicate expected sharing boundaries to be applied by the recipient(s). The four colours are red (named recipients only), amber (limited distribution), green (community-wide distribution), and white (unlimited distribution). GEIGER only receives TLP:WHITE data for now. Figure 2 shows some examples of events shared from CERT-RO to GEIGER.

GEIGER can then use the CERT-RO CTI feed to update its solution. The technical solution used to process incoming MISP data is summarised in Figure 3. Information is exchanged between the GEIGER cloud storage and MISP using an information-sharing channel API. MISP JSON is shared via the information sharing channel API and temporarily stored in a raw data storage. The MISP data are then filtered to extract the information used within the GEIGER solution. The filtered information is stored in a database for processed data. Finally, the GEIGER cloud storage obtains the processed MISP events via a call to the API. One can see that GEIGER additionally returns enriched events to MISP. Although this is a unique and useful feature in the GEIGER solution, we will not discuss it further as it falls outside of the scope of this paper.

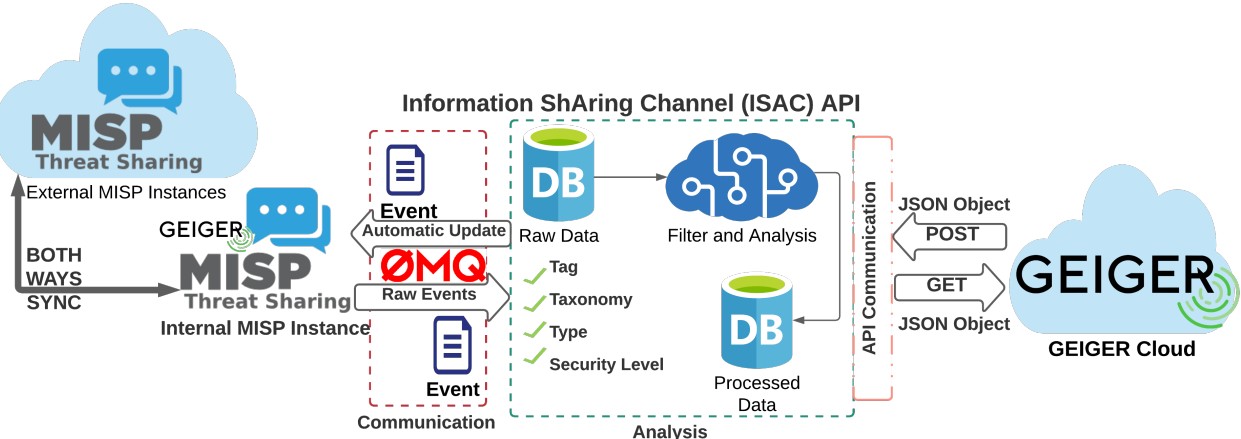

**Figure 3.** Incoming MISP data processed by the GEIGER information sharing channel and stored in the GEIGER cloud storage.

### 3.2. Process: Extracting Insights from MISP Data

In our literature review, we found that researchers are starting to apply supervised machine learning [61], natural language processing (NLP) [47,55], and deep learning [52] techniques to process shared CTI. However, we also found that applying an expert evaluation to the raw data, or using production rules, was far more popular. Of our 47 inclusions, 30 proposed the use of either an expert evaluation or production rules.

This points to the fact that shared CTI often lacks the necessary contextual information for automated reasoning, meaning some form of external knowledge has to be used during processing. This can be in a fully manual process whereby CTI is displayed and it is left to a security expert to decide what to do with the presented data. The other option is to use some form of production rules formulated by security experts a priori, whereby shared CTI can be processed automatically in production.

Of the 11 solutions in our literature review that were relevant to start-ups, digitally dependent SMEs, and digitally-based SMEs, 7 used production rules in their process of turning shared CTI into usable output. This insight led us to conclude that using production rules within the GEIGER solution provides the ideal circumstances to combine expert insights with an automated, usable process for SMEs.

The GEIGER process for using shared CTI from MISP is depicted in Figure 4. We will focus on the threat prioritisation part of Figure 4 here, and discuss recommendation selection and the user interface in Section 3.3. The threat prioritisation process proceeds as follows. First, security experts form a threat classification that is suitable for the SME target group, based on cybersecurity threat reports. How time intensive this task is, depends on whether an existing threat taxonomy can be used, or whether the experts deem an adaptation is necessary for the target group.

In the case of GEIGER, the target audience is primarily the smallest and least digitally mature SMEs. Given their large dependence on external suppliers for IT solutions, we introduced an external environment threat. This category represents threats from third parties and the supply chain. All other threat categories, which can be seen in Figure 1, appear regularly in ENISA's top threat lists. Our experience is that experts with a knowledge of existing threat taxonomies are adept at suggesting an appropriate taxonomy for a specific target group. Additionally, the classification used in the GEIGER project can be used by other projects targeting SMEs, reducing the effort required for classification at these projects. For more details on our classification, see van Haastrecht et al. [88].

Next, the selected threats must be prioritised. We could choose to base prioritisation solely on the shared CTI from MISP. Yet, although MISP's threat intelligence provides a plentiful and continuous stream of data, it does not contain the information that allows us to create distinct prioritisations for different SME categories. As we outlined in our solution requirements at the start of Section 3, SME cybersecurity solutions must recognise the heterogeneous nature of the SME landscape. The GEIGER solution achieves this by

creating different threat prioritisations for digitally dependent SMEs, digitally-based SMEs, and digital enablers. Start-ups are not treated separately, since prioritisation of threats is largely dependent on an enterprise's nature in the digital environment, rather than how long it has been in existence.

Our initial threat prioritisation was constructed based on expert insights regarding information from cybersecurity reports. The expert interpretation is required to ensure that rankings and prioritisations from reports are adequately mapped to the GEIGER use case: SMEs. Additionally, we used the insights from our VCDB analysis. We mentioned the potential issues with using VCDB data in an SME cybersecurity solution in Section 2. However, the analysed data can provide insights into how threat frequencies progressed over time and which threats are especially relevant to particular SME categories. An example of such an observation is that denial of service is less relevant to digitally dependent SMEs than to digitally-based SMEs.

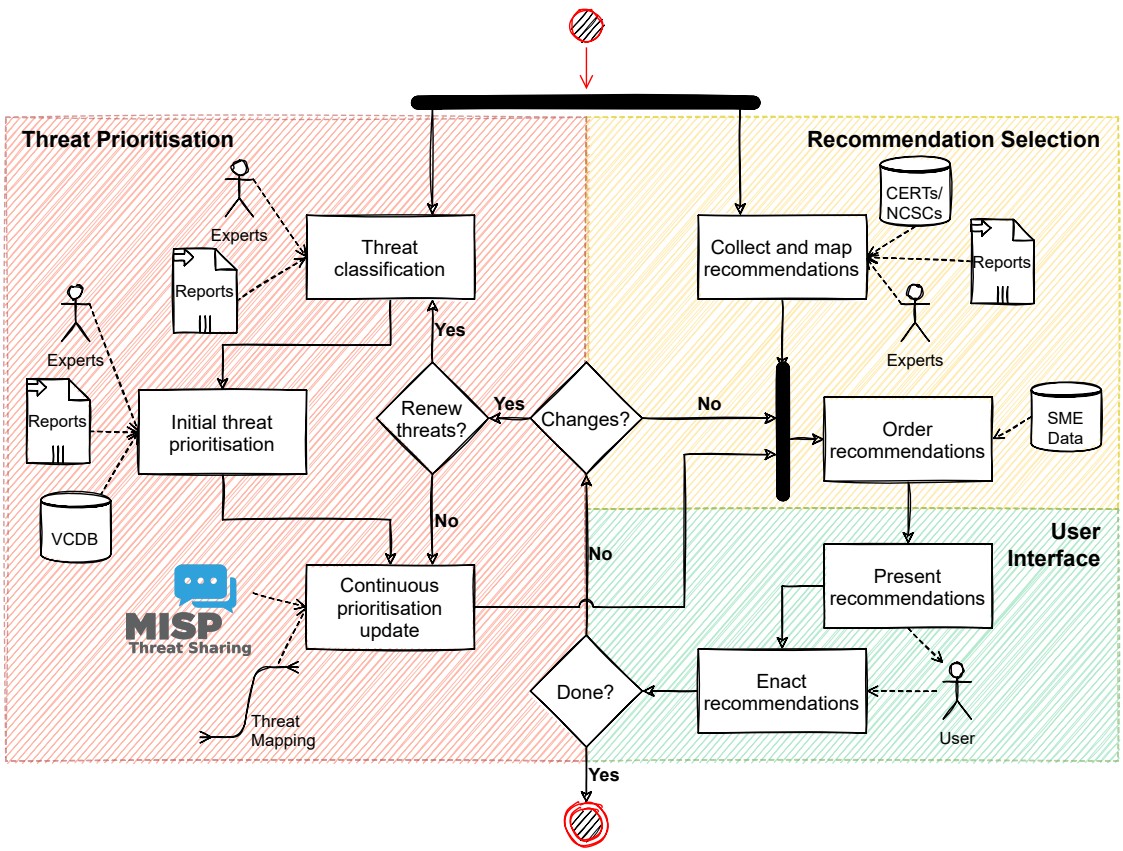

**Figure 4.** Process for turning shared CTI into actionable recommendations for SME users.

We can then use the MISP threat intelligence to continuously update our tailored threat prioritisation. However, CERT-RO's MISP taxonomies do not directly map to our ENISA-derived threat classification. Hence, we first need to use a threat mapping to map the incoming threats to the GEIGER threat classification. This mapping step is also depicted in Figure 3, as 'Filter and Analysis.' We can then apply our production rules to update our threat prioritisation based on the new information we receive.

An example algorithm that can be used to update threat weights is shown in Algorithm 1. In future work, we aim to compare several predictive algorithms to arrive at a conclusion regarding which algorithm is best suited to this context. For now, however, an exponential smoothing algorithm as described in Algorithm 1 can be considered as a baseline.

We define a set of threats $T$ and a set of SME categories $C$. We must know our initial threat weights matrix $\omega$, with dimensions $|T| \times |C|$. The goal of the algorithm is to determine an updated weights matrix $\omega^*$, based on the MISP incident data available. To update

our weights, we use an exponential smoothing approach inspired by the more advanced intermittent demand forecasting approaches known from operations research [92]. In exponential smoothing, new data does not fully determine how we update our forecasts. Instead, we define a smoothing factor $\alpha \in [0,1]$, which determines how much weight the new data receives compared to the data we already have. A lower value of $\alpha$ results in less weight for new data, and therefore a smoother progression over time. The value of $\alpha$ has historically been chosen in the range 0.1–0.2 [93]. In our example algorithm, we use $\alpha = 0.1$, to ensure that weights do not change too much over time.

---

**Algorithm 1** An example of an exponential smoothing algorithm that can be used to update threat weights

---

    **initialise**: $\alpha = 0.1$, $t_{new} = 1$, $t_{old} = 12$
    **obtain**: $\mathbf{n}_{new}$, $\mathbf{n}_{old}$, $\boldsymbol{\omega}$
    **set**: $\boldsymbol{\omega}^* = \boldsymbol{\omega}$

    **for** threat $i$ **in** $T$ **do**
        **if** $n_{old,i} = 0$ **then**                           $\triangleright$ Cannot update when no earlier data
            **continue**
        **else**
            $time\_ratio = \frac{t_{old} - t_{new}}{t_{new}}$
            $\bar{n}_{old,i} = \frac{n_{old,i}}{time\_ratio}$
            $n_{smooth,i} = \alpha \times n_{new,i} + (1 - \alpha) \times \bar{n}_{old,i}$        $\triangleright$ Apply exponential smoothing
            $multiplier = \frac{n_{smooth,i}}{\bar{n}_{old,i}}$
            **for** category $c$ **in** $C$ **do**
                $\omega_{i,c}^* = multiplier \times \omega_{i,c}^*$
            **end for**
        **end if**
    **end for**

    $weight\_sums = [0 \textbf{ for } c \in C]$                     $\triangleright$ Normalise weights to sum to 1
    **for** category $c$ **in** $C$ **do**
        $weight\_sums[c] = \sum_{i \in T} \omega_{i,c}^*$
        **for** threat $i$ **in** $T$ **do**
            $\omega_{i,c}^* = \frac{\omega_{i,c}^*}{weight\_sums[c]}$
        **end for**
    **end for**

    **return**: $\boldsymbol{\omega}^*$                                          $\triangleright$ Return updated weights

---

        The final inputs we must determine are the time intervals that we consider for updating. These intervals determine how often our algorithm should be executed, and thus how often we update threat weights in the GEIGER solution. We elected to update our weights every month, to ensure that we can respond quickly to a changing threat landscape. One might then ask: Why not update every week or every day?

        We have two main reasons for not updating more frequently. First, by updating very frequently we increase the influence incident outliers have on our weights. If on a particular day a large number of malware incidents are shared via MISP, this would lead to an increase in our malware weights, even though this may be unwarranted when looking at a longer period. The second reason is more practical. Users of the GEIGER application will receive recommendations based on our threat prioritisations. If we change our weights daily, users will have to deal with different prioritisations daily. From a user experience perspective, this would not be ideal.

        Hence, we selected to update monthly, making the previous month the period where we consider reported incidents to be new. We label this period as $t_{new}$ and the corresponding

array of incident frequencies per threat $\mathbf{n}_{new}$. Similarly, we introduce $t_{old}$ and $\mathbf{n}_{old}$. For these variables, we choose to look back one year, meaning incidents reported between one month ago and one year ago fall in the 'old' category.

Through the application of our exponential smoothing algorithm as outlined in Algorithm 1, we update our threat weights monthly. By updating our threat prioritisation, we ensure that the information we provide to SMEs accurately represents the current threat landscape. This allows GEIGER users to receive information on what actions they should take to counter the most pressing threats.

The process of threat prioritisation is continual, as the cyber threat landscape is ever-changing. Besides the periodic updates provided by the MISP data, we also periodically assess whether our threat classification and initial threat prioritisation should be updated. As witnessed by the consistency in the ENISA top threats, completely new types of cybersecurity threats do not appear often. Nevertheless, given the dynamic nature of the cyber threat landscape and the constant struggle between cyber attackers and defenders, any cybersecurity solution must have controls in place to deal with major, unexpected shifts. If we observe major changes to the cyber threat landscape in our GEIGER periodic evaluations, we will repeat the complete threat prioritisation process to ensure our prioritisations are as accurate as possible.

### 3.3. Output: Providing Actionable Recommendations

We observed in Section 2 that shared CTI solutions applying an extensive filtering process to arrive at actionable insights, are most suited to the least digitally mature SMEs. Simply providing SMEs with tailored threat prioritisations is not enough if we want to motivate them to take action. Given their lack of internally available cybersecurity expertise and resources, they need to be given clear and actionable instructions, rather than generic advice. The recommendation selection and user interface components of Figure 4, serve the purpose of providing SMEs with the guidance they require.

Our process starts with collecting the latest cybersecurity recommendations—sometimes termed countermeasures—from reports such as those of ENISA and the websites of national CERTs and National Cyber Security Centres (NCSCs). Many of these sources offer advice aimed specifically at SMEs, although expert input is required at this stage to filter out any irrelevant recommendations. As with earlier stages where expert input was required, the time investment is modest and results can be reused by other projects with a similar target audience.

We must then determine which recommendations apply to which threats. Luckily, many sources provide such mappings, making it relatively simple to couple recommendations to threats in the collection phase. Mapping is done with the help of experts. However, this step in the process could be automated in the future, especially if sources providing recommendations start using standardised formats to make their data available. Knowing SME characteristics such as its category, we can then order recommendations based on relevance to the SME. Finally, we can present the ordered recommendations to the user, who can choose to enact the recommendations they deem most relevant. Figure 5 shows how the GEIGER user interface presents recommendations to users.

The user receives prioritised, personalised, and actionable recommendations, without needing to first provide extensive internal data. As with any risk assessment solution, providing more data will help the SME to gain a more accurate picture of the cybersecurity risk they face. However, the user can get started without such data. This makes our approach accessible to start-ups and digitally dependent SMEs, who are in dire need of cybersecurity assistance.

### 3.4. Practical Example

To provide insight into how the process of Figure 4 works in practice, we will cover a practical example in this section. The steps of our example are presented in Figure 6. This example follows the true structure of the GEIGER application using real MISP data. However, the responses of the GEIGER user interface depicted in Figure 6 do not represent

a GEIGER installation at an actual SME, but rather a simulation of how the GEIGER application would respond. In the discussion of Section 4, we will elaborate on our plans for evaluating the application in an operational setting.

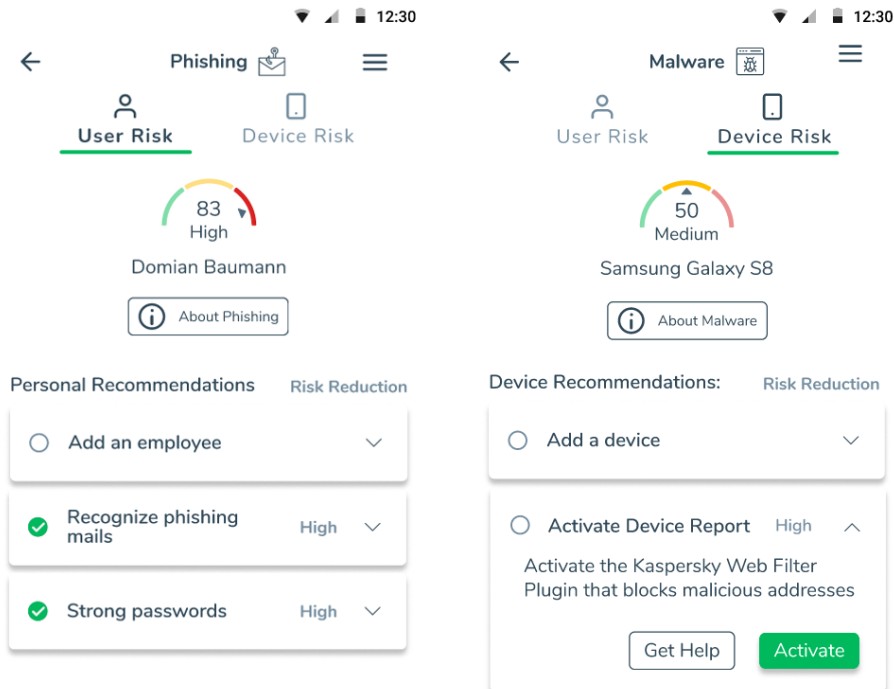

**Figure 5.** Phishing and malware recommendations shown to the user in the GEIGER user interface.

Recently, a malware variety termed 'Flubot' infected Android devices across Europe and Australia [94]. An increased frequency of malware incidents should be reflected in how we prioritise threats for SMEs, given that other threats are not similarly on the rise. Figure 6 explains how our solution would respond to a Flubot malware wave. As the wave hits, Flubot incidents will start to appear in CERT-RO's MISP feed. The feed depicted in Figure 2 would change to include incident descriptions similar to the one shown in Figure 6.

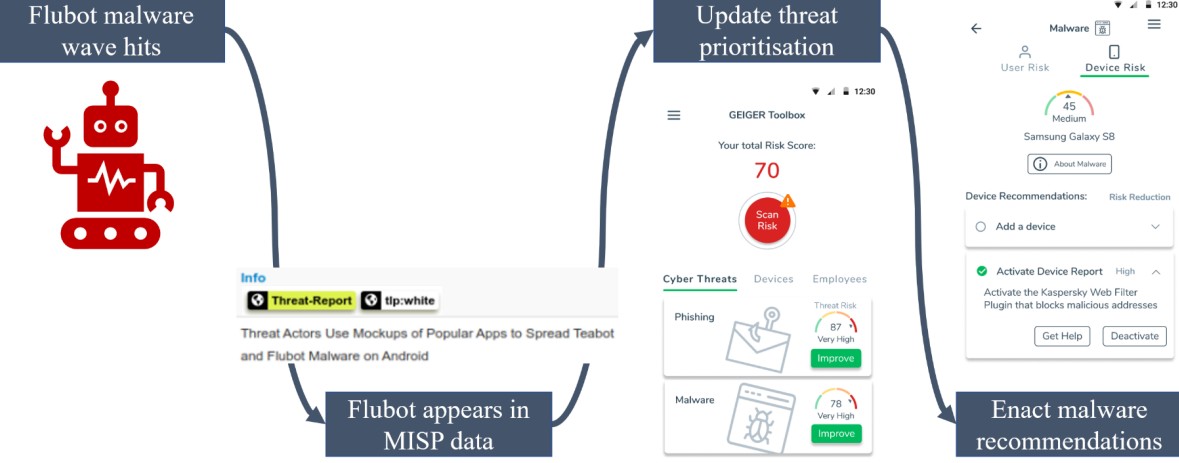

**Figure 6.** Our solution responds to the Flubot malware wave based on incoming MISP data.

The MISP data are then processed further within the GEIGER solution. Figure 3 showed the technical components and interactions involved in filtering MISP data and storing it in the GEIGER cloud storage. The next time the exponential smoothing algorithm of Algorithm 1 is executed, the relatively high incidence of malware will cause the malware

threat type to receive a higher priority. The user will be notified of a change in the prioritisation and can act accordingly. Although recommendations themselves will not be updated, the change in threat prioritisation will motivate the user to enact malware recommendations sooner rather than later.

This example highlights that just because many SMEs do not have the resources to actively monitor the cyber threat landscape, does not mean they are incapable of responding to changes. We need to construct solutions that automate the tasks SMEs are unable to perform while enabling SMEs in the tasks only they can execute. In the end, it is up to the SME to take action and implement recommendations. We, as cybersecurity experts, should do our utmost to ensure SMEs are in a position to act with confidence and determination.

## 4. Discussion

At the outset of this paper, we asked: How can shared incident information be used to help improve SME cybersecurity? Our literature review showed that approaches exist that could be used to help digitally-based SMEs and digital enablers, but that start-ups and digitally dependent SMEs are largely left to their own devices (see Table 2). We methodically analysed our review results and motivated how solutions building on structured external CTI show promise in helping the least digitally mature SMEs. This motivation is one of the main contributions of our paper. Structured open-source intelligence also has potential, but, as our analysis of VCDB demonstrated, is likely to have biases in the data collection phase that are problematic for use in SME solutions.

In Section 3, we introduced three requirements for an SME cybersecurity solution. The GEIGER solution as described in van Haastrecht et al. [88] embedded understandable recommendations collected from CERTs and NCSCs throughout Europe in an intuitive user interface. This ensured that SMEs consider recommendations actionable (Requirement 1). It additionally used input from cybersecurity experts, reports, and VCDB to create a threat prioritisation tailored to an SME's category. Thus, the solution could adapt to different SME characteristics to offer tailored advice (Requirement 2). However, the solution did not yet ensure a timely response to changes in the cyber threat landscape (Requirement 3). The addition of using incoming MISP data to continuously update our threat prioritisation solved this issue.

The use of shared CTI to continuously update threat prioritisations is the key addition to the GEIGER solution presented in this paper. However, it should be evaluated in conjunction with the other components of the GEIGER solution. The automation achieved by incorporating shared CTI is only useful if other parts of our solution facilitate, rather than offset, this automation. We delineated in Sections 3.2 and 3.3 that although expert input is mandated periodically, the time investment required of experts is modest. Additionally, the use of expert input and other data sources allows us to incorporate threat impact as a contributing factor besides threat frequency. Basing our prioritisations purely on MISP incident frequency data would ignore the importance of impact in the risk equation.

Section 3.4 provided a practical example of our solution in a simulated environment and we based our solution on a broad range of existing insights regarding SME cybersecurity. Nevertheless, we may have overseen certain implications of using our application in the real world. It remains a limitation that we have not yet evaluated our solution in an operational environment. We will discuss this limitation and others in the following paragraphs.

*Study Limitations*

Our methodology and solution have their limitations, and our current state of evaluation is one of them. Although our application is currently complete in a prototype components implementation, its impact and relevance remain to be proven in an operational environment. An in-depth investigation of the optimal algorithm choice for updating threat weights is another future necessity. This evaluation is intended to begin once the GEIGER information sharing channel has been operational for several months.

A further limitation is that we focused our literature review on the period since the introduction of MISP in 2016. Although recent years have seen remarkable progress in the shared CTI field, it is certainly possible that we overlooked ideas for suitable solutions by restricting our timeline. Additionally, our solution is dependent on the continued popularity of MISP as an incident sharing platform. MISP facilitates data exchange using the STIX format, which is the de facto standard for information exchange in the cybersecurity field. MISP, however, is not the only standard when it comes to threat sharing platforms. However, we believe in its future given the large support it receives from CERTs throughout Europe.

A final point to mention is that the validity of our solution relies on the inclusion of new cybersecurity threats in CERT-RO's MISP feed. Currently, the threats we include in our solution are all covered by one or more MISP incident types. However, if a new threat appears that is relevant to SMEs, this threat may not be represented in CERT-RO's MISP feed. This could happen if the nature of the threat makes it relevant to SMEs, yet not to CERT-RO. We believe our tight cooperation with CERT-RO and other CERTs throughout Europe offers sufficient potential for mitigation of this risk, but it is present.

## 5. Conclusions

Small- and medium-sized enterprises (SMEs) generally do not have the knowledge and resources to deal with cybersecurity threats. Therefore, we need to assist them in raising their cybersecurity awareness and resilience. A solution is to share the cyber threat intelligence (CTI) of other organisations with SMEs. After all, a problem shared is a problem halved. Yet, shared CTI is rarely used in solutions to address SME needs. Especially the least digitally mature SMEs are often overlooked.

Through reviewing the shared CTI literature, we found potential in structured, externally gathered CTI feeds to aid the most vulnerable SMEs. Our solution incorporates an external CTI feed to continuously update threat prioritisations for SMEs. By mapping publicly available countermeasure suggestions to our prioritised threats, we can provide SMEs with actionable recommendations ordered by relevance.

We tailored our threat prioritisations to SME characteristics to recognise the heterogeneous SME landscape. Different SME categories deserve different treatment due to, for example, varying amounts of internal cybersecurity data being available. Our solution does not place a heavy burden on SMEs to provide internal data, making it ideally suited to less digitally mature SMEs.

In future, we will continue to develop our solution and seek to employ it in operational environments. We intend to perform a thorough analysis to determine improved variants of our current threat updating algorithm. We must keep investing time in evaluating our solution with SMEs to ensure it aligns with their needs. The eventual goal is an environment where shared CTI aids all organisations, no matter their cybersecurity maturity.

**Author Contributions:** Conceptualisation, M.v.H., R.K., C.P., S.F., J.F.R., M.B. and M.S.; methodology, M.v.H., M.B. and M.S.; software, M.v.H., L.B., J.F.R. and E.A.; validation, M.v.H., M.B. and M.S.; formal analysis, M.v.H.; data curation, M.v.H.; writing—original draft preparation, M.v.H., G.G., G.T., R.K., C.P., D.D. and A.R.; writing—review and editing, M.v.H., G.G., G.T., R.K., C.P., D.D., A.R., M.B. and M.S.; visualisation, M.v.H., L.B., J.F.R. and E.A.; project administration, M.v.H. All authors have read and agreed to the published version of the manuscript.

**Funding:** This work was made possible with funding from the European Union's Horizon 2020 research and innovation programme, under grant agreement No. 883588 (GEIGER). The opinions expressed and arguments employed herein do not necessarily reflect the official views of the funding body.

**Institutional Review Board Statement:** Not applicable.

**Informed Consent Statement:** Not applicable.

**Data Availability Statement:** The code of our algorithm and the data resulting from our VCDB incident analysis are publicly available here: https://github.com/cyber-geiger/cloud-logic. The code

of other GEIGER components can be found here: https://github.com/cyber-geiger, accessed on 9 November 2021.

**Conflicts of Interest:** The authors declare no conflict of interest. The funders had no role in the design of the study; in the collection, analyses, or interpretation of data; in the writing of the manuscript, or in the decision to publish the results.

## Abbreviations

The following abbreviations are used in this manuscript:

| | |
|---|---|
| CERT | Computer Emergency Response Team |
| CIRCL | Computer Incident Response Center Luxembourg |
| CTI | Cyber Threat Intelligence |
| DBIR | Data Breach Investigations Report |
| ENISA | European Union Agency for Cybersecurity |
| IoC | Indicator of Compromise |
| MISP | Malware Information Sharing Platform |
| NCSC | National Cyber Security Centre |
| NCSP | National Cyber Security Platform |
| NLP | Natural Language Processing |
| SMEs | Small- and Medium-Sized Enterprises |
| SOC | Security Operations Centre |
| STIX | Structured Threat Information eXpression |
| TAXII | Trusted Automated eXchange of Indicator Information |
| TLP | Traffic Light Protocol |
| VCDB | VERIS Community Database |
| VERIS | Vocabulary for Event Recording and Incident Sharing |

## Appendix A. Threat Definitions

**Table A1.** Definitions of threats, based on ENISA [87], NIST [95], and van Haastrecht et al. [88] definitions.

| Threat | Definition |
|---|---|
| Botnets | "A network of connected devices infected by bot malware" [87]. |
| Data breach | "A data breach is a type of cybersecurity incident in which information (or part of an information system) is accessed without the right authorisation, typically with malicious intent, leading to the potential loss or misuse of that information" [87]. |
| Denial of service | "The prevention of authorised access to resources or the delaying of time-critical operations" [95]. |
| External environment threats | Threats of financial, reputational, or legal damages due to noncompliance with regulations, standards, or other agreements with third parties. Includes the threats resulting from changing financial and economic circumstances and the actions (intended or unintended) of external stakeholders such as customers and suppliers. [88] |
| Insider threats | The potential of "an entity with authorised access" to "harm an information system or enterprise through destruction, disclosure, modification of data, and/or denial of service" [95]. |
| Malware | Short for malicious software. Malware is any program written with the intent to carry out "harmful actions" [95]. |
| Phishing | "Phishing is the fraudulent attempt to steal user data such as login credentials, credit card information, or even money using social engineering techniques" [87]. |
| Physical threats | Threats related to the "tampering, damage, theft, and loss" of physical assets [87]. |
| Ransomware | Ransomware is a type of malware "that infects the computer systems of users and manipulates the infected system in a way that the victim cannot (partially or fully) use it and the data stored on it" [87]. The victim is pressed to pay a ransom to regain access. |
| Spam | "The abuse of electronic messaging systems to indiscriminately send unsolicited bulk messages" [95]. "It is considered a cybersecurity threat when used as an attack vector to distribute or enable other threats" [87]. |
| Web application threats | Threats to the security of web applications and services, often abusing misconfigurations, weaknesses, or vulnerabilities in the implementation of these applications [87]. |
| Web-based threats | Web-based threats are "an attractive method by which threat actors can delude victims using web systems and services as the threat vector" [87]. |

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
