# Peer review of "A Shared Cyber Threat Intelligence Solution for SMEs"

_electronics, doi:10.3390/electronics10232913_

Round 1

Reviewer 1 Report

The paper is well structured and written. However, there remain a few things for the authors to consider:

  • Please discuss alternatives to SYMBALS, or at least detail why it was selected as a methodology

  • Figures need to be rearranged, as they seem to overflow text bounds

  • Line 208 - why the focus on companies with <=100 employees, if the common cutoff for an SME is 250 employees?

  • Figure 2 includes the name of at least 1 company operating in Romania (OLX) - this should be changed or anonymized

  • Increase the resolution of Figure 3 and make sure its translated to English (Taxonomie)

  • Line 380 - value of alpha parameter seems to be arbitrary, or at least there is no thorough explanation provided for the selected value

  • The last paragraph reads less like a research paper and more like an advertisement.

Author Response

Dear reviewer,

We thank you for taking the time to review our submission and for your kind comments. We will proceed to cover each of your remarks, pointing to the areas of the text we have edited.

Thank you for your remark regarding the choice for the SYMBALS methodology. We agree that a more extensive discussion of our motivations is beneficial to the paper. We added a couple of paragraphs on our motivations in lines 81-96.

Regarding your comments on the figures overflowing text bounds, we have evaluated all our images and made sure none violate the MDPI stipulations. One thing to note is that MDPI offers a ‘widefigure’ format whereby figures and tables are allowed to be wider than the text width.

We clarified our choice for using the 100-employee cut-off for VCDB in lines 222-227. The reason for using this cut-off point is that the next cut-off point available in VCDB is at 1,000 employees. We weighed the choice for using both cut-offs and ended up using the 100-employee cut-off point, also since the focus in our solution is on the less mature SMEs.

Thank you for your sharp remark regarding Figure 2. We blurred the company name to anonymize the figure appropriately. Additionally, we used a sharper resolution version of Figure 3 and ensured to replace the incorrect spelling of taxonomy.

You make a fair point regarding the choice for the alpha parameter. The reason for choosing this value is that this is a standard value in the literature and has been since the introduction of the method by Croston (1972). We added this explanation to the text in lines 403-408.

Regarding your final comment, we consider this a harsh but fair judgement. We have altered the final paragraph of our conclusion to a humbler rendition.

Your comments, which were at times precise and at times deep, showed a clear understanding of our ideas. We hope to have adequately addressed these comments. Regardless, we thank you for the time spent in reviewing our submission.

Yours,

The Authors

Reviewer 2 Report

The paper provided an extensive overview of the existing cyber threat intelligent sharing solutions for SMEs and pointed out that the current solutions do not necessary cover the needs of the least digitally mature SMEs. The authors proposed a solution that collects and filters data from the MISP incident sharing platform, prioritises the threats, and provide SMEs with actionable recommendations. The authors also presented a prototype application integrating the proposed solution. The paper is well written and easy to follow. I recommend the authors to improve the paper by considering the following points:

  1. In my understanding, the algorithm 1 prioritises threats based on only incident frequency. I wonder why other important factors especially threat impact (severity) is not considered. I suggest authors including a discussion on this point.
  2. Figure 4: please some explanations about the experts’ tasks for initial threat prioritisation, threat classification, and collect map recommendations. What knowledge/algorithm/analysis need to be applied by each of these experts, how many years of expertise is needed for the expect to do each of these tasks? How much effort is needed for them (hours/days per new record? continuous working) to complete one task? Is it possible to automatise some of these experts’ tasks?
  3. Please add short descriptions (e.g., a table) about the different threats that appear in Figure 1.
  4. In P7, please give different examples of taxonomies that can be used to classify objects of MISP data.  
  5. In P5, please mention, which kind of data (database) does ENISA use to classify data breaches?  
  6. In P9, line #339: I suggest rephrasing the statement “an external environment threat representing threats from …”

Author Response

Dear reviewer,

Thank you for your kind comments and your suggestions for improvement. We have attempted to address all the points you have raised, which we will walk through one-by-one in the following paragraphs.

Regarding your point on updating only based on incident frequency, this is correct. Although updating by impact would be ideal, this data is not as readily shared in current shared CTI platforms. However, we do include impact in our initial threat prioritisations, since many of the reports (e.g., ENISA) we use include this dimension. We have delineated this incorporation of impact in our edited discussion (lines 530-533).

We agree that more explanation was required on the exact role expert input played in creating the GEIGER solution. We have interweaved information on how expert input was incorporated in our discussions in lines 354-386 (threat prioritisation) and lines 452-462 (recommendations). We hope this helps to clarify this point.

You make a good point that a description of each of the threats we discuss is a good addition. Since this table became quite long, we have included it in the appendix of the paper (Appendix A, Table 3, line 604).

We have added two examples of taxonomies which MISP supports (lines 299-300). We additionally added a description of the ENISA threat landscape report data collection process (lines 204-205), as per your request. Finally, we agree that the statement originally on line 339 was somewhat confusing. We rephrased the sentence, which can now be found on lines 360-361.

We want to thank you once again for taking the time to review our submission. Your comments helped us to improve our paper considerably. We hope that you consider our efforts to have adequately addressed your comments.

Yours,

The Authors

Reviewer 3 Report

The paper looks very interesting and addresses a highly demanded problem (cybersecurity of SMEs with limited resources). Conducting a systematic literature review as part of the paper is very good too. However, I have the following comments.

----------------------------------

The abstract did not show how the prototype application is evaluated. 

At line 57, it was written "Based on what we know of current trends in cybersecurity intelligence sharing literature ...". What you know is not measurable nor not supported by references. 

The problem of why SMEs need a special type of cybersecurity solution is not clearly stated. This seems the main problem of the study, so it should be clearly stated in one paragraph before writing the RQ at line 64. 

In Line 71, it was written ""We then describe our proposed solution using the Malware Information Sharing Platform (MISP) [14] in Section 3,....." it is confusing that you propose MISP while it is already published in [14].

The contribution of the paper should be given toward the end of the introduction section. 

For section 2, you claimed that you have done a systematic literature review which is fine but you are expected to give more details about the research methodology (what are the inclusion and exclusion criteria, what is the aim objectives, how did you validate the findings of this literature review). Then it is expected to classify the literature summarised in section 2. Then show what is missing, what is the gap in the current literature? what lessons were learned from conducting this systematic literature review, how this is connected to your current study?

In line 253, it was written " We summarise our three requirements for an
SME cybersecurity solution below:" it should be clear how you came up to these requirements. there is no any reference or your own study to support these requirements.

In section 3 (Shared CTI Solution for SMEs), you spent much space talking about the existed solution (GEIGER [86]) without making it clear what is the weakness of the GEIGER, what you have added to GEIGER to improve it. It would be clear if you first give an overview of GEIGER and identify its limitations.  Then it is expected that your current study to address these limitations. This should be your "proposed solution".  This is the main section of the paper. It should be clear for any reader such that they can replicate the study when reading it. 

It is not clear whether the proposed solution is evaluated or not. In section,"3.4.", the authors wrote about  "Practical example" but it is not clear whether this is conducted in an SME or not. If yes, this should be used as a case study. However, you still need to explain in details how this was conducted, what are the size of the company, what are the objectives to measure/achieve during the study? what type of threats/attacks be exposed to? have all attacks/threats been detected? 

The discussion section is rather ad hoc. The discussion should discuss and analyse the findings of the experimental or simulation results of the proposed system. It also should discuss how well the objectives of the system have been achieved as well as provide a comparison with the literature especially (GEIGER [86]). I can not see these in the current discussion. 

English showed be carefully proofread. There are many small paragraphs of only two sentences. 

Figures' quality should be improved. Figure 2 can be read. 

Author Response

Dear reviewer,

We thank you for your kind words and for acknowledging the relevance of the problem we are considering. Of course, we equally thank you for your comments and suggestions for improvement. We will proceed by covering each of your comments and pointing to the areas in the text where we have made changes to address the comments.

You are correct to point out that our abstract did not discuss how our prototype was evaluated. We have edited the abstract (lines 14-16) to better reflect this part of our research.

Thank you for your sharp observation regarding what we wrote on line 57. We have edited the sentence in question, which can now be found on line 58.

Although we had discussed the need of SMEs for a specialised cybersecurity solution in the introduction already, we have extended this discussion based on your comments. The initial motivation can now be found in lines 24-29 (edited) and lines 48-62 in the introduction. We return to some of these points in the literature review, especially based on the findings in Van Haastrecht et al. (2021, reference 7). We also added references when discussing our solution requirements in Section 3, but we will return to those in addressing one of your further comments.

We agree with the reviewer that the sentence in line 71 was somewhat confusing in relation to MISP. It was not our intention to ‘propose’ MISP, simply to propose a solution, which uses MISP. We have changed the sentence to reflect this better, which can now be found in lines 74-75.

We additionally understand your comment that a mention of our paper contributions at the end of the introduction is appropriate. We have edited the final paragraphs of the introduction (lines 67-79) to explicitly mention our contributions.

Thank you also for your comment regarding our literature review. We have added more details on our procedure and why we have chosen this specific methodology in lines 83-96. Regarding your further comments on classifying the literature, this is what we have done in Table 2, where we classify all inclusions based on two dimensions: the organisation type it is suitable for and the type of CTI it uses. We discuss our review results at length in the remainder of Section 2. We hope you consider all these contributions together as a sufficient detailing of our literature review.

 We understand your remarks regarding the requirements we present at the start of Section 3. We have clarified the text, while also making explicit reference to the sources we used to motivate our requirements. The changes can be found in lines 260-273.

Your comments related to the earlier version of the GEIGER solution relate to a similar section of the paper. We agree more explanation was necessary of why GEIGER needed to be adapted and what we have done to improve it. We first extended our GEIGER introduction, now to be found in lines 254-259. We also added an extensive discussion on this topic in our edited discussion section, in lines 514-533.

We regret that our description of evaluation was unclear in the original manuscript. We have extended section 3.4 to describe our steps more clearly (lines 474-480). We also addressed the current shortcomings of our evaluation in more detail in the limitations mentioned in lines 539-544.

Regarding the discussion, you mention that our original discussion section came across as ad hoc. We have restructured our discussion to place more focus on how the objectives of our paper were achieved and how our proposed adaptations to GEIGER improve the GEIGER solution. The first paragraph and last two paragraphs of our discussion are largely the same, but the content in lines 514-544 are new. We hope to have addressed some of your concerns regarding the discussion with these changes.

Finally, we turn to your grammatical and stylistic comments. We have proofread the English of our paper. The main alteration we made is to merge short paragraphs that essentially belonged together, to solve the problem of “many small paragraphs” which you mentioned. Regarding our figures, we reviewed all figures for legibility and improved the quality of Figures 3 and 6. Since you mention Figure 2 can be read, the only change we made there is to anonymize a company name which showed up in the original figure.

To summarise, we want to thank you once more for the time you have invested in reviewing our submission. Your comments demonstrated a clear understanding of our ideas and process, which we truly value. We hope to have adequately addressed your comments, and certainly feel your comments have helped to improve our manuscript.

Yours,

The Authors

Round 2

Reviewer 3 Report

Thanks for addressing the comments. It is clear that the authors put good effort into improving the paper.  The paper looks much better now. 

One minor thing which could further improve the paper presentation. I do understand the proposed solution has not been evaluated. This was stated in line 537 "For now, it remains a limitation that we have not yet evaluated our solution in an operational environment".

I would suggest using the lines from 537 to 544 and making it clear in a sub-section of Section 4 and you may name it "study limitations".  

Best wishes

Author Response

Dear reviewer,

Thank you for your kind words and your further suggestion for improvement. We agree that it improves the paper to clearly delineate the limitations discussion from the remainder of Section 4 (Discussion).

We have added a 'Study limitations' subsection as per your suggestion. You can find the subsection in lines 540-562. We have kept the original line 537 as an introductory sentence to lead into the subsection. We have also chosen to include the other limitations we mention in the 'Study limitations' subsection, which is why the section spans the lines 540-562, rather than lines 540-546.

We hope these changes adequately address your comment. Thank you once more for helping us to improve our submission.

Yours,

The Authors